# Diet Modifications towards Restoration of Insulin Sensitivity and Daily Insulin Fluctuations in Diabetes

Ana Magalhães [1,†], Cátia Barra [1,2,*,†], Ana Borges [1,3] and Lèlita Santos [1,2,4]

1 Internal Medicine Service, Coimbra Hospital and University Centre, 3000-548 Coimbra, Portugal
2 Coimbra Institute of Clinical and Biomedical Research (iCBR), Faculty of Medicine and Center of Innovative Biomedicine and Biotechnology (CIBB), University of Coimbra, 3000-548 Coimbra, Portugal
3 Faculty of Medicine, University of Coimbra, 3000-548 Coimbra, Portugal
4 CIMAGO Research Centre, Faculty of Medicine, University of Coimbra, 3000-548 Coimbra, Portugal
* Correspondence: 12071@chuc.min-saude.pt
† These authors contributed equally to this work.

**Abstract:** The circadian rhythm is essential in order to maintain metabolic homeostasis and insulin sensitivity. Disruption of circadian mechanisms is associated with the development of metabolic diseases, such as diabetes. Lifestyle changes such as an equilibrated diet and physical activity are known to improve glycaemic control in diabetic patients. One of the mechanisms possibly involved in such an improvement is the restoration of insulin circadian rhythms. There are several available dietary schemes based on circadian rhythms. Some of them are associated with better regulation of daily insulin fluctuations and the improvement of Type 2 Diabetes and metabolic syndrome. In the current review, we aim to explore how the different types of diet can impact glucose metabolism and insulin sensitivity in patients with diabetes, highlighting the interactions with the mechanisms of circadian insulin rhythm and the prevention of hyperinsulinemia.

**Keywords:** daily insulin fluctuation; insulin sensitivity; diets; circadian rhythm





## 1. Introduction

Insulin was discovered by Frederick Banting and Charles Best in 1921. It is a pancreatic beta-cell anabolic hormone initially produced by as pre-insulin and then as proinsulin after a maturation process [1]. Subsequently, it is translocated from the endoplasmic reticulum to the Golgi apparatus, where it is cleaved into C-peptide and insulin, which are simultaneously released by exocytosis [2]. Its production by pancreatic beta cells is enhanced in response to glucose, and it is responsible for maintaining constant blood levels of glucose and metabolic homeostasis. However, it has been demonstrated that amino acids and fatty acids are also capable of stimulating insulin secretion [3,4]. Its secretion from pancreatic beta cells is biphasic: the first phase occurs with a rapid release, while the second phase is characterized by a more sustained and less elevated release [5]. Insulin levels are increased in the postprandial period when the circulating glucose levels increase, making the first phase very significant. In this phase, insulin inhibits glucagon secretion from the pancreatic alpha cells through paracrine action at the beginning of the meal. On the other hand, the second phase is critical for maintaining this inhibition and increasing glucose storage and utilization [6]. Thirty minutes after the beginning of the meal, insulin suppresses lipolysis in the adipose tissue by inactivating the hormone-sensitive lipase, leading to a decrease in non-esterified fatty acids (NEFA) and glycerol in the blood [7,8]. The lower blood NEFA and higher insulin levels induce the suppression of glucose production in the liver and an increase in glucose utilization (glycolysis) by the muscle [3,9,10]. Insulin and glucagon secretion, as well as glucose homeostasis, are also regulated by the incretins, which are secreted by the gut in response to food intake [11,12]. Glucagon-like peptide-1 (GLP-1) and glucose-dependent insulinotropic polypeptide (GIP)

are the two major incretins derived from gut neuroendocrine cells and are involved in these mechanisms, and their receptors may be found both in alpha and beta cells [11,13].

Insulin secretion dysregulation due to several causes promotes the development of diabetes. Type 1 diabetes (T1D) is characterized by auto-immune destruction of beta cells, while type 2 diabetes (T2D) is related to the development of insulin resistance, which is associated with obesity and metabolic syndrome. In T2D, beta-cell destruction is also observed through their progressive exhaustion and dysfunction in a later phase [14,15]. Consequently, beta-cell hyperplasia and a hyperinsulinemic state are often observed in obese patients in order to compensate for chronic low-grade inflammation and insulin resistance [14,16,17]. Moreover, hyperinsulinemia itself contributes to insulin resistance due to insulin receptor downregulation [11]. Lifestyle changes such as an equilibrated diet and physical exercise may prevent these events and thus contribute to maintaining a normal weight and insulin sensitivity in the peripheral tissues.

## 2. Circadian Rhythm Influences Insulin Secretion

The circadian rhythm is essential for maintaining a normal body physiology. It is mediated by specific components that are controlled by the daily light–dark and feeding–fasting cycles. The central nervous system plays an important role in the circadian rhythm through synapse networking, which acts in different brain regions and in endocrine cells, contributing to the circadian regulation of metabolism [18]. These outputs are integrated into the hypothalamus, which is responsible for hunger–satiety, thermoregulation, sleep–arousal, and osmolarity [19].

The metabolic activity of several organs is also mediated by their internal clocks and circadian rhythms. In the intestine, the expression of sodium-glucose transport protein 1 (SGLT-1) has a rhythmic cycle, which is increased when the glucose intake is anticipated. Similar mechanisms regulate insulin secretion. Besides being regulated by glucose levels and incretins, its exocytosis also has a circadian regulation, possibly because incretins also have a circadian rhythm themselves [20]. Some studies in healthy human subjects have shown daily fluctuations in glucose tolerance, suggesting that it is greatest in the morning, while a significant decrease occurs during the night [21,22]. This is not only because of the oscillations in peripheral insulin sensitivity, but also the variations of glucose-stimulated insulin secretion during the daily 24 h period [23].

Uncoupling protein 2 (UCP2) has been described to influence metabolism, as it is expressed by endogenous circadian oscillators in pancreatic islets and acts in beta cells as a negative regulator of glucose-stimulated insulin secretion (GSIS). As mentioned before, insulin regulates the metabolic response to fasting and its suppression is crucial to allow for endogenous glucose production in the liver and lipolysis in the adipose tissue in order to release metabolic fuel. Therefore, Ucp2/UCP2 has been suggested to coordinate insulin secretion according to nutrient ingestion, so an upregulation of Ucp2/UCP2 prevents hypoglycaemia during the fasting period by inhibiting insulin secretion and promoting fuel mobilization [24,25]. This type of regulation influences the activity of several other endocrine pathways such as melatonin, glucocorticoids, and growth hormone (GH) [20]. GH and cortisol secretion are regulated by their own circadian rhythm, and are higher in the nocturnal period, which contributes to insulin resistance in the early morning hours [22].

It has been demonstrated that chronic disruption of these circadian mechanisms is related to the development of metabolic diseases, as established by several studies in shift workers [26,27]. In a retrospective study conducted by Garaulet and her collaborators, eating early (lunch time before 15:00 p.m.) results in enhanced weight loss effectiveness, suggesting a relation between eating intervals and the day–night cycle in the metabolism [28]. Other studies have revealed that time-restricted feeding is capable of improving metabolic diseases due to sustained diurnal rhythms and imposing daily feeding–fasting cycles [29,30]. Similar to what happens with GH and cortisol, melatonin (Figure 1) also promotes a hyperglycaemic state in the evening, given that melatonin binding to its receptor in pancreatic islets inhibits insulin release from the beta cells [31]. Reciprocal interactions between

metabolism and the circadian clock imply that nutrition quality, quantity, and daily eating patterns can affect diurnal rhythms, which in turn determine whole-body physiology.

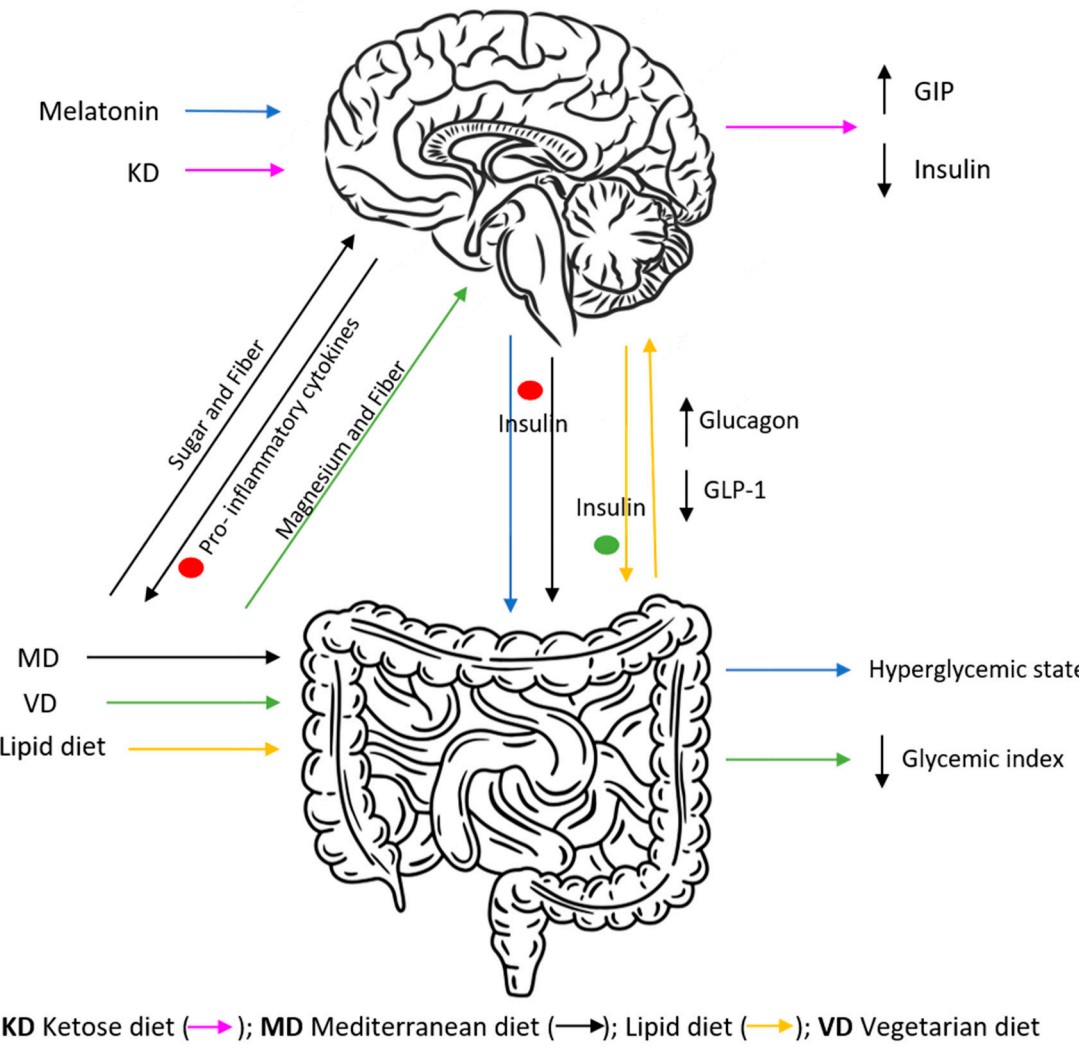

**Figure 1.** Effects of different diets on insulin and incretins secretion, as well as hyperglycaemic and inflammatory states. KD, characterized by the ingestion of <50 g of carbohydrates/day, promotes a decrease in insulin and increase in glucagon and GIP fasting levels. MD, because of its higher fiber supply and reduced sugar ingestion, is capable of decreasing blood insulin levels and increasing GIP secretion, similarly decreasing pro-inflammatory cytokine activation and reducing the risk of diabetes development. VD, based on reduced calorie ingestion and being rich in magnesium and fibre, promotes a decrease in GI. A lipid diet is associated with diabetes development by increasing glucagon secretion and decreasing GLP-1.

## 3. The Impact of Different Diets on the Levels of Insulin Secretion

### 3.1. The Impact of Diet

Nowadays, most diets that are called healthy can be included in a few categories: they match the Mediterranean Diet, or they are low in fat, low in carbohydrates (CH), or are vegetarian [32]. Nutrition is considered one of the pillars for the prevention and treatment of several diseases, namely metabolic syndrome [33]. Besides the amount of food ingested, growing attention has been devoted to the quality of macronutrients and the role they play in the regulation of T2D [34,35]. The current discussion about food standards suggests that it is not just a question of quantity, but also of quality and, importantly, time and schedule. For example, eating most calories and carbs at lunchtime or in the early afternoon, avoiding

late dinners, and keeping the number of meals and the timing of meals consistent, all play a role in regulating postprandial blood glucose and insulin sensitivity [36].

This section of the review will focus on postprandial insulin, and the long-term insulin response in relation to the glycaemic index (GI), the area under the curve (AUC) of the glucose response after CH consumption [37].

### 3.2. Mediterranean Diet

The Mediterranean Diet emerged in the 1960s as the diet with the most beneficial impact on health and the one recommended by experts. This diet is characterized mainly by the consumption of vegetables, with the regular consumption of protein (meat, fish, or egg), CH, and a limited amount of fats, usually from olive oil. Nonetheless, it is a diet low in fat and rich in vitamins with antioxidant and anti-inflammatory properties [35]. Therefore, as it decreases the activation of the pro-inflammatory cytokines (Figure 1), it also reduces chronic inflammation and the risk of T2D. As it includes a low consumption of sugar/CH, it also has a role in regulating diabetes. So, it is suggested that by resulting in a greater supply of fibre and less sugar, it is effective at decreasing the load of this component on the body, while also decreasing the circulating insulin levels [38].

### 3.3. Vegetarian Diet

Adherence to a vegetarian diet is growing worldwide. This is characterized by the exclusion of foods of an animal origin. However, if this diet is not supplemented, macro and micronutrient deficiencies such as omega-3 fatty acids, iron, vitamin D, calcium, zinc, iodine, and vitamin B12 may occur [32]. The reinforced intake of vegetables and cereals reduces the risk of T2D as they are rich in magnesium and fibre (Figure 1). The insulin signalling pathway is magnesium-dependent and fibre helps to lower the GI [39]. The low GI of this diet may explain the lower cardiovascular risk and greater insulin sensitivity compared with omnivores [37].

Within the scope of T2D, a plant-based diet will reduce body weight, as it reduces the ingested calories, and increases postprandial metabolism [34]. Some studies suggest that vegetarians have a lower amount of intramuscular lipids than omnivores, and this proportion also occurs at a hepatic level. By having this metabolic (catabolism) effect, it will decrease HbA1c and postprandial lipids. Visceral fat reduction also decreases the levels of plasma inflammatory adipokines and oxidative stress markers, which may have an effect on beta cell function and insulin regulation. An unexpected effect of this diet is that it increases the thermic effect of food, supporting this theory of catabolism [40]. The question that arises in this diet is its suitability for the entire population as it is a difficult diet to maintain, at least for the Western population, as a high intake of CH has to be monitored by a nutritionist [32].

### 3.4. Paleolithic Diet

This diet is based on the consumption of fruits, vegetables, eggs, fish, and meat, with the latter as a source of protein. Thus, it excludes all processed food, including dairy products. Thus, it contains less CH but a greater protein intake [41]. In fact, the distribution of nutrients occurs as follows: 35% lipids, 35% CH, and 30% protein [42]. The Paleolithic diet causes a reduction in insulin concentration after a meal, that is, less insulin is needed with this nutrient distribution than with other diets [43]. Studies have suggested that this approach decreases insulin secretion in the long-term and increases beta cell sensitivity to postprandial glucose load [42]. However, Genoni et al. suggest that this adaptation may not be beneficial in terms of intestinal motility [44]. Biologically, a catabolic state is stimulated and postprandial glucagon suppression is observed. Together with insulin secretion, this may result from improved postprandial secretion of the incretins GLP-1 and GIP into circulation. Their increase also promotes satiety and thus weight loss [41].

## 4. Diets That Promote Ketosis

This group of diets is divided into three types: low in CH, normoprotein, and high in lipids, such as the ketogenic diet. It also includes the famous «fasting» and calorie restriction, which increase ketosis due to the lack of calories. Despite a restriction of nutrients, some studies have considered this type of diet more healthy as it helps to preserve muscle mass, increase weight loss, reduce appetite, and above all decrease insulin resistance and circulating insulin levels [45,46].

### 4.1. Ketogenic Diet

It consists of a normocaloric diet well known for rapid weight loss. However, in terms of insulin patterns and its effect on the liver, it is still controversial [47]. The proposal is to ingest less than 50 g of CH per day, which will mimic a catabolic state. Insulin secretion will decrease as the main nutrient will be ketone bodies (from lipids) as an alternative substrate [45]. Fasting glucagon and GIP levels increase, while, after ingestion, postprandial glucose and insulin AUC are lower than in a control meal (Figure 1). The increased adiponectin observed after this diet may also promote insulin sensitivity [46].

### 4.2. Lipid-Modified Diet

As would be expected, this diet may increase the risk of obesity and insulin resistance [48], but some studies claim it is a matter of quantity. The type of ingested lipids affects the insulin signalling cascade. Contrary to expectations, intervention studies have reported that replacing monounsaturated with saturated lipids alters the plasma fatty acids, but may increase insulin sensitivity [34]. However, some studies report that this diet causes insulin resistance. A 2-week lipid-rich diet leads to insulin resistance in muscle tissue due to metabolic changes, specifically in the mitochondria (incomplete oxidation of molecules), and impaired signalling of the insulin cascade [49]. Given that this type of diet causes obesity, this may have other consequences, such as a decreased secretion of GLP-1 (Figure 1) [41]. Thus, the ratio between insulin and glucagon production in the pancreas is changed, leading to an increased appetite and increased caloric intake, which will cause an overstimulation of insulin and induce resistance [41].

### 4.3. Low Carb Diet (LCD)

This diet is often used by nutritionists as it promotes weight loss in a short period of time and, consequently, an improvement in the glycaemic profile. This diet is based on the consumption of vegetables and protein, removing virtually all CH (consuming < 130 g/day, the main foods originating glucose) [50]. In this context, the mathematical formula is simple: reducing the intake of glucose (present in CH) reduces hyperinsulinemia, which can be adapted for patients with diabetes [51]. There is evidence that this diet allows 93% of patients to regress to pre-diabetes, 46% of patients no longer need medication, and 60% of patients regress from diabetes within 1 year [51]. However, it has to be noted that a low glycaemic diet (GD) is not a low GI diet [52]. LCD improves the glycaemic profile because it rapidly increases postprandial insulin levels. However, a high glycaemic peak will cause patients to have less satiety and avoid a further increase in dietary intake [46].

### 4.4. Diet Rich in Carbohydrates

Controversial studies report that the ingestion of a diet rich in CH and fibre can lead to normoglycemia in prediabetic individuals [53]. After eating a meal, insulin in fat cells stimulates the entry of glucose into the tissues, decreasing the release of lipids and inhibiting the production of ketone bodies. Thus, this diet will produce hyperinsulinemia that will force the deposition of lipids instead of their oxidation [54]. The mechanisms are not clear; there are studies reporting that a diet rich in CH reduces blood glucose through glucose metabolism at the muscle level, improving the glycaemic profile in the short and long term [55]. This occurs if CH is not refined, with an increased adiponectin concentration, a decrease in body weight, and thus an improvement in the metabolic profile [56]. However,

it is not clear what would be the long-term effects on beta cell function, because their overstimulation is expected to increase the risk of exhaustion.

Once again, nutrients are paramount, and diets with a GI < 55 are digested slowly and those with a high GI (>70) are digested quickly [52]. Importantly, high GI foods can alter leptin by interfering with satiety beyond the absorption spectrum [57]. So, foods that contain CH, such as vegetables and fruit, have better GI values, satisfying appetite for a longer time and improving the glycaemic profile [52].

*4.5. Caloric Restriction*

This dietary intervention consists of reducing the caloric intake by 25–30% from the baseline, but without restricting the necessary nutrients, that is, decreasing the quantity without decreasing the quality [45]. This diet can exacerbate some metabolic problems as it can increase appetite and stress hormones [54]. In this catabolic context, glucagon is favored and the secretion of GLP-1 is inhibited [41]. As a consequence, the hepatic glucose output is increased. However, pancreatic apoptosis decreases and insulin sensitivity increases, mainly due to the lower insulinemia [45]. It is especially problematic in patients with T2D, possibly causing hypoglycaemia and difficulties in the long-term regulation of glucose metabolism. Caloric restriction is more suitable for patients in the early stages of metabolic dysregulation, where it will favour body weight and fat mass loss, while decreasing lipotoxicity in the muscle, liver, and adipose tissue. This will decrease low-grade inflammation and increase insulin sensitivity. Thus, caloric restriction is suitable to prevent insulin resistance and beta-cell exhaustion in patients with obesity or prediabetes, although it is a dietary scheme that is very hard to follow by patients with T2D (reviewed by Joaquim et al., 2022 [58]). For patients with T2D, other less severe schemes are recommended, such as time-restricting fasting (TRF). In this dietary approach, patients are recommended to eat only during one part of the day, not consuming any calories during the other part, usually for at least 10 h. This approach often does not require caloric restriction, only fasting for half of the day in order to regulate relevant biological mechanisms such as circadian rhythms, stress hormones suppression, and autophagy (reviewed by Joaquim et al., 2022 [58]). As mentioned above, dietary consumption at specific times may coordinate the response to the presence or absence of nutrients, namely their gut metabolization and absorption; insulin secretion and action; and all the necessary changes in the liver, muscle, and adipose tissue metabolism. TRF has been shown to promote overnight mobilization of energy reserves through lipolysis, lipogenesis, and hepatic glucose production, contributing to a higher catabolic activity [59–61]. The rhythmic regulation of SGLT1 in the gut and UCP2 in beta cells, as already discussed, may in fact explain the positive effects of this type of diet on the beta-cell function and insulin sensitivity. It is possible that the overnight suppression of insulin may be a key factor in better diurnal insulin sensitivity and preserved beta-cell function (reviewed by Joaquim et al., 2022 [58]).

**5. Conclusions**

In this review, we focus on daily insulin fluctuation and glucose metabolism, as well as the influence of diet on these mechanisms. Metabolic homeostasis is highly coordinated by the circadian rhythm, which regulates insulin and glucagon secretion in order to maintain a normal fuel balance between tissues. However, it is well known that disruption of these mechanisms can lead to metabolic disease development, such as T2D. Current dietary habits are known to cause beta-cell overstimulation due to the high consumption of CH, especially refined ones. This will not only cause long-term beta-cell exhaustion but also insulin resistance, as hyperinsulinemia is one of the major drivers of insulin receptor inactivation. Nowadays, there are several diets known to improve insulin sensitivity and better control glycaemic profile in patients with T2D. Postprandial hyperglycaemia depends mainly on meal composition, but other factors also contribute to its magnitude, such as glucose absorption; secretion; and the action of incretin hormones, insulin, and glucagon. Thus, it is important to evaluate the quality of the nutrients rather than only their quantity,

and, most importantly, their glycaemic impact. In T2D, it is important to adjust the diet to control glycemia after meals and the postprandial and fasting insulin levels.

**Author Contributions:** Conceptualization: A.M., C.B., A.B. and L.S. Methodology: A.M. and C.B.; A.B. and L.S. Writing—original draft preparation: A.M. and C.B. Writing-review and editing: A.B. and L.S.; Supervision: A.B. and L.S. All authors have read and agreed to the published version of the manuscript.

**Funding:** This work was supported by Foundation for Science and Technology (FCT), Portugal (Pest UID/NEU/04539/2013 and UID/NEU/04539/2019: CNC.IBILI; Pest UIDB/04539/2020 and UIDP/04539/2020: CIBB).

**Institutional Review Board Statement:** Not applicable.

**Informed Consent Statement:** Not applicable.

**Data Availability Statement:** The data presented in this study are available on request from the corresponding author.

**Acknowledgments:** Cátia Barra is a CIBB member, who is supported by the Portuguese Science and Technology Foundation (FCT), Strategic Projects UIDB/04539/2020 and UIDBP/4539/2020.

**Conflicts of Interest:** The authors declare no conflict of interest.

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
