# Peer review of "Diet Modifications towards Restoration of Insulin Sensitivity and Daily Insulin Fluctuations in Diabetes"

_diabetology, doi:10.3390/diabetology3040046_

Round 1
Reviewer 1 Report
I thank the journal for giving me an opportunity to review the paper titled “Diet modifications towards restoration of insulin sensitivity and daily insulin fluctuations in diabetes” by Magalhaes et al. The paper presents information on how different types of diets can help mitigate insulin sensitivity and its fluctuations specifically in individuals with T2D. Overall, the review has relevance but one of the main criticisms is the lack of connectivity and flow between the different subtopics that are discussed. The language also requires ample corrections throughout the manuscript. As per the authors, one of the main aspects of this paper is the role of circadian rhythm (the abstract mentions it) during the regulation or dysregulation of metabolic homeostasis, and how the review “highlights the interactions with the mechanisms of circadian insulin rhythm and the prevention of hyperinsulinemia”. The title of the paper does not bear any reflection to this whatsoever. Section 2 in the paper mentions the role of circadian rhythms influencing insulin secretion but there is no connectivity or crosstalk with the subsequent sections. Even the conclusions are completely devoid of this key aspect that the authors mentioned in the abstract itself (why not rephrase the title as well to include the aspect of circadian rhythms?). While the section on different diets impacting levels of insulin secretion is overall informative, the authors are requested to better stitch the manuscript to cover all the essential topics together. Major revisions are required in this review before it can be considered again for publication. Some other corrections to be considered are as follows -
1. Language corrections are needed. For example, line 11 start of the abstract itself: metabolic homeostasis ≠ insulin sensitivity. The sentence reads as if metabolic homeostasis means insulin sensitivity.
2. Line 49, please rephrase to “auto-immune” destruction of beta cells
3. Line 52, language error – sentence requires correction (In T2D is also observed that….)
4. Line 53 – incorrect language
5. Lines 97-98 requires language change
6. Figure 1 needs to be organized better – “citokines” spelling (cytokines); hiperglycemic spelling (hyperglycemic). Figure 1 requires reference source if it has been modified from some article/source.
7. Line 132 (we know that…): how do we know this? Reference?
8. Line 217 – what is “GH”/”LCD” diet? Not expanded anywhere in text. What is HC (line 219)?
9. Please make sure ALL abbreviations are expanded in text as it becomes very difficult for a reader to follow what the authors are meaning to write.
10. The conclusions are poorly written, where is the inclusion of the concept of ‘circadian rhythm’ and how it ties into the topics being discussed (its role in diet modification and mitigating insulin sensitivity).
Author Response
RESPONSE TO REVIEWERS
We want to thank the editor and the reviewers for their critical assessment of the manuscript. We have acted upon the recommendations of the reviewers in order to improve the manuscript quality. Below we added our point-by-point responses (in blue text) to each comment (in black text). In addition, we highlighted revisions in yellow in the updated manuscript.
REVIEWER 1:
Comments and Suggestions for Authors:
I thank the journal for giving me an opportunity to review the paper titled “Diet modifications towards restoration of insulin sensitivity and daily insulin fluctuations in diabetes” by Magalhaes et al. The paper presents information on how different types of diets can help mitigate insulin sensitivity and its fluctuations specifically in individuals with T2D. Overall, the review has relevance but one of the main criticisms is the lack of connectivity and flow between the different subtopics that are discussed. The language also requires ample corrections throughout the manuscript. As per the authors, one of the main aspects of this paper is the role of circadian rhythm (the abstract mentions it) during the regulation or dysregulation of metabolic homeostasis, and how the review “highlights the interactions with the mechanisms of circadian insulin rhythm and the prevention of hyperinsulinemia”. The title of the paper does not bear any reflection to this whatsoever. Section 2 in the paper mentions the role of circadian rhythms influencing insulin secretion but there is no connectivity or crosstalk with the subsequent sections. Even the conclusions are completely devoid of this key aspect that the authors mentioned in the abstract itself (why not rephrase the title as well to include the aspect of circadian rhythms?). While the section on different diets impacting levels of insulin secretion is overall informative, the authors are requested to better stitch the manuscript to cover all the essential topics together. Major revisions are required in this review before it can be considered again for publication. Some other corrections to be considered are as follows -
We are grateful to the reviewer for the positive evaluation of our manuscript and for the careful review. Regarding the title, it already mentioned the importance circadian rhythms on insulin fluctuations because we already mentioned “restoration of daily insulin fluctuations”. However, we agree that the integration between diet and circadian rhythms could be improved and we changed that during this revision. We highlighted in yellow all the changes made in the manuscript. We thank the reviewer for the comments. Please find below a point-by-point response to all the comments, questions and suggestions raised.
- Language corrections are needed. For example, line 11 start of the abstract itself: metabolic homeostasis ≠ insulin sensitivity. The sentence reads as if metabolic homeostasis means insulin sensitivity.
We corrected the language.
- Line 49, please rephrase to “auto-immune” destruction of beta cells
We corrected the typo.
- Line 52, language error – sentence requires correction (In T2D is also observed that….)
We corrected the sentence.
- Line 53 – incorrect language
We corrected the language.
- Lines 97-98 requires language change
We corrected the language.
- Figure 1 needs to be organized better – “citokines” spelling (cytokines); hiperglycemic spelling (hyperglycemic). Figure 1 requires reference source if it has been modified from some article/source.
We corrected all the typos.
- Line 132 (we know that…): how do we know this? Reference?
We added the reference in this sentence of the manuscript, which was the same of the previous sentences. We were more careful with this type of sentence without references.
- Line 217 – what is “GH”/”LCD” diet? Not expanded anywhere in the text. What is HC (line 219)?
We expanded these abbreviations in the manuscript.
- Please make sure ALL abbreviations are expanded in the text as it becomes very difficult for a reader to follow what the authors are meaning to write.
We adjusted all abbreviations of the manuscript.
- The conclusions are poorly written, where is the inclusion of the concept of ‘circadian rhythm’ and how it ties into the topics being discussed (its role in diet modification and mitigating insulin sensitivity).
Thank you for the suggestion. We tried to improve that information and connection throughout the manuscript.
Reviewer 2 Report
The authors have explained very clearly about various diets and their impact on insulin sensitivity and circadian rhythm of insulin secretion.
Minor comments:
1. Figure Legend: Please explain the figure in detail for better understanding.
2. Figure 1: pro-inflammatory “Citocines” should be “cytokines”; “Hiperglycemic” should be “Hyperglycemic”.
3. Can the authors expand the following abbreviations:
Line 63: CNS
Line 132: MD
Line 217: GH
Line 217: LCD
Line 219: HCs
4. Can the authors add reference to the following statements:
a. Line 191- 193: “Contrary to expectations, intervention studies…….”
b. Line 200 - 201: “In obese individuals there is……”
c. Line 214 – 216: “There is evidence that………”
Author Response
RESPONSE TO REVIEWERS
We want to thank the editor and the reviewers for their critical assessment of the manuscript. We have acted upon the recommendations of the reviewers to improve the manuscript's quality. Below we added our point-by-point responses (in blue text) to each comment (in black text). In addition, we highlighted revisions in yellow in the updated manuscript.
REVIEWER 2:
Comments and Suggestions for Authors: The authors have explained very clearly about various diets and their impact on insulin sensitivity and the circadian rhythm of insulin secretion.
We are grateful to the reviewer for the positive evaluation of our manuscript and the careful review. Please find below a point-by-point response to all the comments, questions and suggestions raised.
- Figure Legend: Please explain the figure in detail for better understanding.
Thank you for the comment, we improved the figure legend.
- Figure 1: pro-inflammatory “Citocines” should be “cytokines”; “Hiperglycemic” should be “Hyperglycemic”.
Thank you for the careful review. We corrected the typos in the manuscript.
- Can the authors expand the following abbreviations: Line 63: CNS; Line 132: MD; Line 217: GH; Line 217: LCD; Line 219: HCs
Thank you for the careful review. We corrected all abbreviations in the manuscript.
- Can the authors add a reference to the following statements:
- Line 191- 193: “Contrary to expectations, intervention studies…….”
- Line 200 - 201: “In obese individuals there is……”
- Line 214 – 216: “There is evidence that………”
Thank you for the careful review. We added references in these sentences of the manuscript.
Round 2
Reviewer 1 Report
Many thanks for considering the review comments. While the review has improved considerably, extensive English language editing is still required for the entire manuscript. Please make extensive language/grammar corrections before this manuscript is accepted for publication (Attached are examples of some comments to revise the manuscript).

Author Response
REVIEWER 1:
Many thanks for considering the review comments. While the review has improved considerably, extensive English language editing is still required for the entire manuscript. Please make extensive language/grammar corrections before this manuscript is accepted for publication (Attached are examples of some comments to revise the manuscript).
We are grateful to the reviewer for the positive evaluation of our previous revisions. We made several language corrections during this second revision and tried to improve the manuscript to the best of our knowledge. We highlighted in yellow all the changes made in the manuscript.
